# Characteristics of Horizontal Precipitation in Semi-Humid Forestland in Northern China

**Jianbo Jia** [1] , **Wende Yan** [2,3,*], **Xiaoyong Chen** [2,3] **and Wenna Liu** [4]

1   College of Forestry, Central South University of Forestry and Technology, Changsha 410004, China; jotham880303@163.com
2   College of Life Science and Technology, Central South University of Forestry and Technology, Changsha 410004, China; XChen@govst.edu
3   National Engineering Laboratory for Applied Forest Ecological Technology in Southern China, Changsha 410004, China
4   College of Soil and Water Conservation, Beijing Forestry University, Beijing 100083, China; lwnfdxl@163.com
*   Correspondence: csfuywd@hotmail.com; Tel.: +86-187-740-093-09

**Abstract:** Little information is available on horizontal precipitation in forest land in semi-humid climate regions. In this study, the quantity and duration of horizontal precipitation were investigated using the high precision weighing lysimeter system in the mountainous areas of northern China during the experiment year 2011 and 2012. The purpose of this study was to better understand the formation mechanisms of horizontal precipitation in the semi-humid climate region. The results showed that hourly values of horizontal precipitation distributed between 0 and 0.1 mm, and that the one-night values distributed between 0.2 and 0.4 mm. The number of days with horizontal precipitation accounted for about 45% of the whole year. The average monthly amount of horizontal precipitation was 4.5 mm in the non-growing season, while it was a mere 1.6 mm in the growing season. The total amount of horizontal precipitation in the year was about 33 mm. Horizontal precipitation represented about 4.61% and 4.23% of the annual precipitation in 2011 and 2012. During the non-growing season, water vapor absorbed by the soil was greater than canopy and soil condensation, not only in terms of frequency, but also in the cumulated quantity. On a typical day, the canopy and soil condensation was 0.07 mm, accounting for 31.81% of total quantity of horizontal precipitation (0.22 mm). Air temperature, soil temperature and wind speed were negatively correlated with the quantity and duration of horizontal precipitation. This research could provide information for a better understanding of the ecological significance of horizontal precipitation in the semi-humid climate region in northern China.

**Keywords:** horizontal precipitation; lysimeter; semi-humid; meteorological factors; forestland

## 1. Introduction

Horizontal precipitation, such as dew, fog and frost, is water vapor that is suspended over the earth's near-surface [1]. When the wind blows through plant and soil surfaces, small droplets can be intercepted by these surfaces, becoming a water source for plants [2]. Horizontal precipitation usually accounts for a small quantity of the total water input in an ecosystem [3]. However, it has been proven to be an important water source in making up the deficit in precipitation, especially in arid environments [4–7]. Furthermore, horizontal precipitation can also improve the development of biological soil crusts [8] and microorganisms [9], increase soil moisture, reduce the vegetation surface temperature and transpiration, which has effects on heat preservation for forest canopies [10–12]. Therefore, horizontal precipitation is an indispensable part of the heat balance, energy balance and water balance in ecosystems.

Like the development of global climate changing, the precipitation process and pattern of forest ecosystem have shown significant variation. Mountainous areas of northern China have been suffering from a gradual decrease in annual precipitation since 1950 [13], while the ecological water consumption in these areas, i.e., evapotranspiration and rainfall, have increased dramatically over the last few decades [14]. At the same time, numerous studies have shown that the forest was not limited by water conditions in such regions [15,16]. Therefore, it is important to understand how to explain the mechanism by which the water relationship between precipitation input and ecological water use is different to what it was. Water balance is the starting point to comprehend the hydrological relationship of forest ecosystem; this must have changed in this region. There may be other water input, which makes up for the lack of precipitation and groundwater. Like the change in water balance, the water cycle of this region might also be different [17]. Horizontal precipitation, which exists nearly all year round, may be the extra water source, and it may have an important effect on the hydrological cycle of forest land in the mountainous areas of northern China.

Current research on horizontal precipitation concentrates on the quantity and duration in different climate types. Plants control the hydrological cycle, but human activities also play a role [18,19]. Hence, different ecosystems may demonstrate different horizontal precipitation. The quantity of horizontal precipitation in different ecosystems is shown in Table 1. And the data came from other literatures that have been cited in Table 1. For forestland, much research focus on arid and humid climate regions. However, little information is available on horizontal precipitation in forest land in semi-humid climate regions such as the mountainous region of northern China.

**Table 1.** Horizontal precipitation in different ecosystems.

| Type | Research Site | Mean Day Value (mm/d) | Mean Year Value (mm) | Literature |
|---|---|---|---|---|
| City | Forest land (Guangzhou China) | 0.034 | | (Ye, 2007) |
| | Industrial district (Guangzhou China) | 0.022 | | |
| | Business district (Guangzhou China) | 0.013 | | |
| | Residential district (Guangzhou China) | 0.009 | | |
| Farmland | Corn field (lowa state USA) | 0.01–0.6 | | (Erik D, 2009) |
| | Soybean field (lowa state USA) | 0.003–0.8 | | |
| | paddy field (Los Banos Philippines) | 0.041–0.218 | | |
| island | Ajaccio island France | 0.036–0.07 | 8.4–9.8 | (Beysens, 2005) |
| | Polynesia island France | 0.068 | 5.58 | (Clus, 2008) |
| | Dalmatian beach Croatia | 0.001–0.592 | 9.3–20 | (Muselli, 2009) |
| desert | Kothara desert India | 0.022–0.25 | | (Sharan, 2007) |
| | Lanzhou China | 0.22–0.25 | | (Li, 2002) |
| Tropic Zone | Xishuangbanna China | 1.36 | 89.4 | (Liu, 2006; 2007) |

When monitoring the horizontal precipitation, there are two additional problems. The first is that the forest canopy structure leads to a complex distribution of horizontal precipitation [20,21]; as a result,

general methods are not able to accurately measure the quantity of horizontal precipitation. The second is that that the water on the surfaces of leaves comes from two sources: the first is water vapor from the air, which is a form of horizontal precipitation because it is input water; the other is plants spitting water, which is not. However, several methods fail to distinguish between these two sources by collecting water samples from leaf surfaces [22]. Due to limitations in measurements and analysis methods, it was difficult to accurately estimate the duration and the quantity of horizontal precipitation, especially in order to evaluate long-term changes [23]. Using different types of materials and instruments, many methods have been developed to measure dew. While all these direct measurement methods were easy to implement, they were not accurate; the reason for this was that their surface properties were different from natural surfaces. Hence, while these dew measurement methods were useful for comparisons, they could not provide an actual quantity value and are were unable to measure water vapor adsorption by soil. In order to solve these problems, a high precision weighing lysimeter system was introduced [3]. This method had four advantages: (1) It could differentiate between the water input from air and that from water spit by the plant, since water spit by the plant did not change the weight of the lysimeter; (2) The weighing precision was high enough to measure the quantity of water input accurately; (3) This method was not affected by the complex forest canopy structure nor by the consequential complex distribution of horizontal precipitation; and (4) it was a long-term and fixed point observation that ensured continuous measurements could be taken throughout the entire research period.

Given the water relationship between decreasing precipitation input and increasing ecological water utilized in the mountainous area in northern China, it is necessary to understand the characteristics and the process of horizontal precipitation in this region. Based on this, we explored: (1) the horizontal precipitation duration and quantity in different forest lands, (2) distinguished quantitative variations in canopy and soil condensation and water vapor absorption by top soil, (3) analyzed the relationship between the formation of horizontal precipitation and influencing factors. Our goal was to provide base information in order to better understand the ecological significance of horizontal precipitation in the mountainous area in northern China.

## 2. Methods

### 2.1. Theory

Water vapor adsorption by soil is an interfacial phenomenon in which gas is adsorbed by a solid. A quantitative discussion of the adsorption of gases on solids was given by Irving Langmuir [24]. The equation relating the quantity of gas adsorbed on a surface to the pressure of the gas at a constant temperature was defined as the adsorption isotherm, which is derived from the kinetic characteristics of the condensation and evaporation of gas molecules at the surface. Langmuir isotherm is based on the gradual coverage of a surface with adsorbed molecules, with saturation occurring when the adsorbed layer has a uniform thickness of one molecule [25].

Soil water mainly exists in one of two forms: attached to the surface of soil particles, or presenting in soil pores. When vapor concentration in soil is different from that in the atmosphere, it will form the concentration gradient between atmospheric and soil levels. Agam and Berliner [2] have shown that when the relative humidity of soil pores is lower than that of air, water vapor from the air can be directly absorbed by the soil, even though the air temperature is higher than the dew point temperature, because a gradient of water vapor concentration exists that could hold water within the soil matrix by adsorption to particle surfaces and/or by capillarity in the pores. These are the two mechanisms which control soil water retention. However, the soil's reaction is influenced by these mechanisms. In particular, water attracted by reactive clay minerals (i.e., adsorption) will cause swelling, but when water is attracted by capillarity into the pores of the soil, swelling does not occur. Temperature and water potential have been suggested as the criteria to determine which of the two mechanisms predominates: when the surface temperature is lower than the dew temperature, the retention curve

is determined by capillarity condensation, whereas when the surface temperature is higher than the dew temperature and the water potential of soil is lower than that of air, it is determined by physical adsorption [26]. No matter which mechanism occurs, a gradient of water vapor concentration from air to soil exists. In order to describe the gradient of water vapor concentration from air to soil pores, the vapor pressure of water in the air above the canopy ($e_a$) and the water potential should be measured. $e_a$ is calculated as follows [27]:

$$e_a = \frac{VPD}{1-RH} = 0.611 e^{\left[\frac{17.502T}{T} + 240.97\right]} \tag{1}$$

where $VPD$ is the vapor pressure deficit, kPa; $RH$ is the soil relative humidity, %; and $T$ is the Kelvin temperature, K. Top soil layer water potential ($\varphi_s$, MPa) was measured using soil water sensors. Air water potential ($\varphi_a$, MPa) is calculated as follows:

$$\varphi_a = 4.6248 \times 10^5 \times T \times lnRH \tag{2}$$

For water vapor absorption by soil, a lysimeter cannot be used directly because it is not able to directly distinguish weight increases, on account of canopy and soil condensation or water vapor absorption by soil. Therefore, we can calculate the water vapor absorption of soil using the following equation:

$$W_s = \int_{t_1}^{t_2} \left(-D\frac{\partial \rho}{\partial x}\right) dt \tag{3}$$

where $W_s$ is the water vapor absorption by soil, g/m²·h; $t_1$ and $t_2$ are the starting and closing times. $\frac{\partial \rho}{\partial x}$ is the gradient of water vapor concentration, which can be obtained from the table of vapor density; and $D$ is the diffusion coefficient, which can be calculated by:

$$D = 1/3 \times \sqrt{\frac{8RT}{\pi\mu}} \times \frac{KT}{\sqrt{2}\pi d^2 e_a} \tag{4}$$

where $R$ is gas constant, 8.31 J/(K·mol); $T$ is the Kelvin temperature. $K$ is Boltzmann constant, $1.38 \times 10^{-23}$ J/K; $e_a$ is vapor pressure of water in the air; $d$ is the diameter of water molecules, $1.925 \times 10^{-9}$ m; $\mu$ is molecular weight, 18.016.

*2.2. Study Site Description*

This study was performed in Jiu Feng National Forestry Park Haidian District, about 30 km north-west of Beijing, China (39°54′ N, 116°28′ E) (Figure 1). It is semi-humid region and 140 m above mean sea level along with 10–25° sloping terrain. Mean daily temperatures range between 23-28 °C from May to mid-October months, and between −5–25 °C throughout winter, with a mean annual temperature of 12 °C. The mean annual precipitation is 630 mm and maximum rainfall occurs between June and September. In the study area, *Pinus tabulaeformis*, *Quercus variabilis*, and *Platycladus orientalis* are the dominant tree species (Figure 2). Most of these species were planted in the 1950s and 1960s. *Lolium perenne,* which is the main undergrowth vegetation (mean 70% ground cover), is widely distributed in the study area. The main shrubs were *Grewia biloba*, *Vitex negundo*, *Broussonetia papyrifera*, whose density was 3–5 plants/m². The soil in the study area is cinnamon soil with an average slope of approximately 10% in a northeastern aspect. The average wind speed is 1.8m/s to 3m/s, and northwest and southeast wind was dominant. Solar exposure and vegetation type maps have been presented by Shi [28].

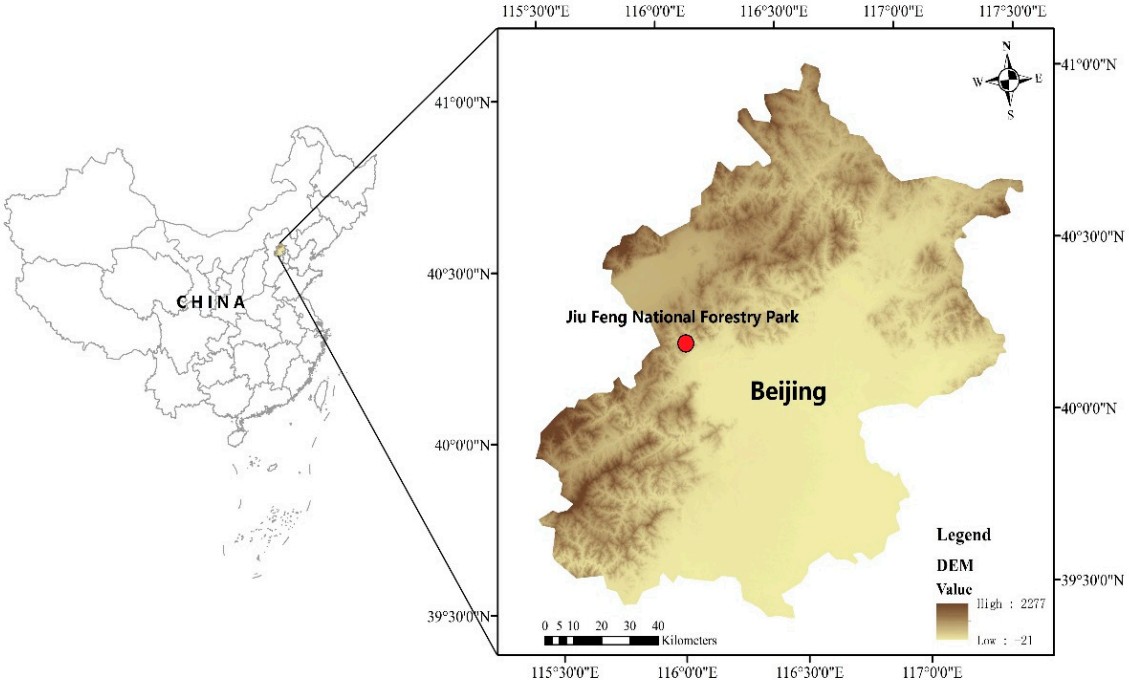

**Figure 1.** The location of Jiu Feng national Forestry Park in Beijing, China (Arc GIS10.0).

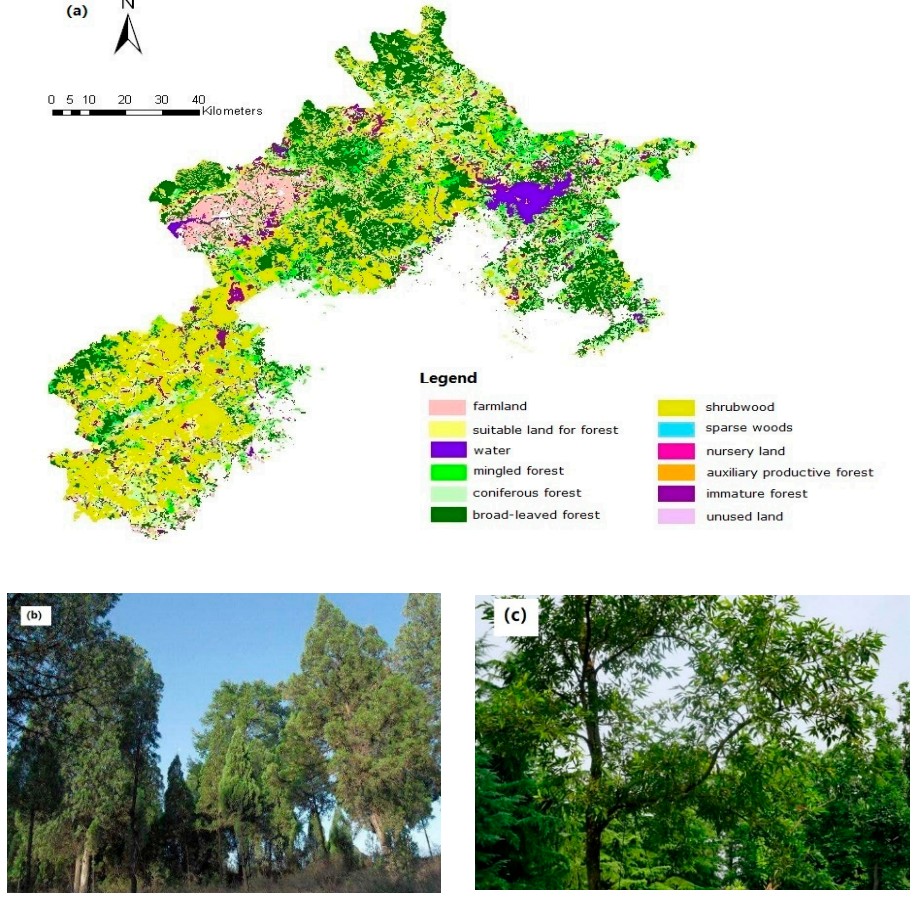

**Figure 2.** *Cont*.

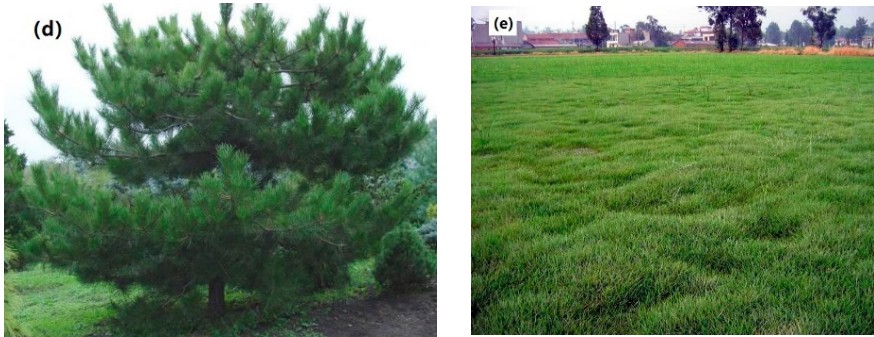

**Figure 2.** Map with vegetation types (**a**) and forest land photos ((**b**) *Platycladus orientalis*; (**c**) *Quercus variabilis*; (**d**) *Pinus tabulaeformis*; (**e**) grass (*Lolium perenne*)).

## *2.3. Horizontal Precipitation Measurements*

The total experimental period was 24 months (from January 2011 to December 2012). Four identical weighing lysimeters (L07W, BiShui, China) were used in this study (Figures 3 and 4). Each lysimeter, filled with undisturbed soil, had a surface area of 4 m$^2$ (2 × 2 m), and a depth of 2.3 m, and could measure mass changes as small as 40 g. This weight corresponded to a water depth of 0.01 mm for the 4 m$^2$ surface areas. This precision was sufficient to determine variations in horizontal precipitation. Four lysimeters were arranged in a rectangle containing grass (*Lolium perenne*), *Platycladus orientalis*, *Pinus tabulaeformis* and *Quercus variabilis* specimens (Table 2). Each tree had nearly the same height (about 10 m) and crown breadth; the trees were all 11 years old. During the monitoring period, the grass was mowed as to maintain the height (mean 5 cm) and density (mean 70% ground cover) within a narrow range. In addition, the same trees were planted around each lysimeter in order to form forest habitat conditions; the planting density was approximately 2500 trees/hm$^2$. There were no other trees around the grass.

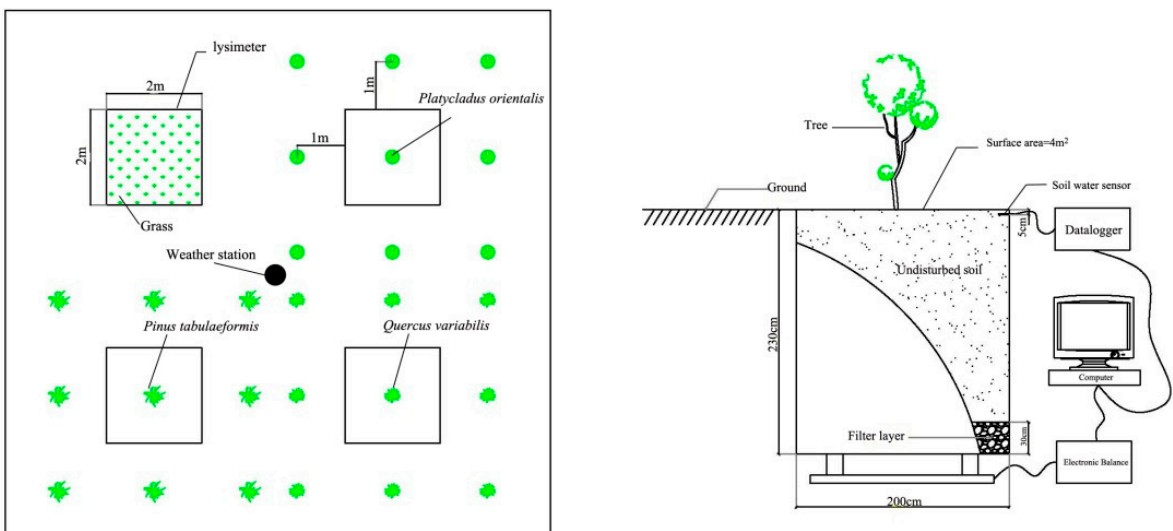

**Figure 3.** The layout situation and profile of lysimeter station.

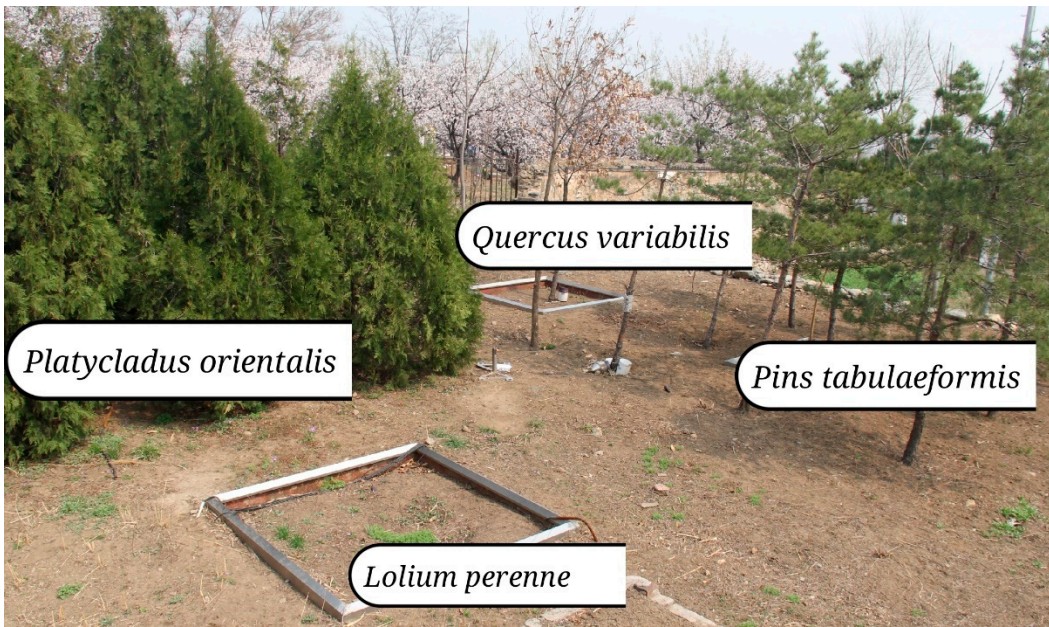

**Figure 4.** The layout situation of four lysimeters (photo by 2012.4.11).

**Table 2.** Vegetation properties in lysimeter system.

| Tree Species | Density (Tree/hm) | DBH * (cm) | Height (m) | Crown Diameter (m) (Length × Width) |
|---|---|---|---|---|
| *Lolium perenne* | ±70% (coverage) | | ±0.05 | |
| *Q. variabilis* | 2500 | ±4.9 | ±2.5 | ±1.2 × 1.3 |
| *P. orientalis* | 2500 | ±5.2 | ±2.5 | ±1.0 × 1.3 |
| *P. tabulaeformis* | 2500 | ±5.5 | ±2.4 | ±1.3 × 1.5 |

* Diameter at breast height.

The lysimeter measured the weight of the soil on the electronic balance and pressure sensors using the lever principle. The weight change of the lysimeter was considered due to the change in the quantity of water input and output based on the following water balance equation:

$$P_v + P_h = ET + R + Q \pm \Delta W \tag{5}$$

where $P_v$ is the quantity of rain and snow, $P_h$ is the quantity of horizontal precipitation, $ET$ is the quantity of evapotranspiration, $R$ is the quantity of runoff, $Q$ is the quantity of deep percolation, and $\Delta W$ is the change in weight in the lysimeter. The $P_v$ value was collected at the site by a continuously recording tipping-bucket rain gauge (RG3-M onset USA) which was installed in the top of meteorological station (U30, Onset, USA) located in the center of the lysimeter. During the process of horizontal precipitation formation, there was no rainfall or snow input. Mass increases of the lysimeter, without rain or snow, were finally classified as horizontal precipitation. Therefore, the value of $P_v$ and $Q$ were all zero. Furthermore, the lysimeter station was installed at level ground and the gradient of the slope was zero, so the value of $R$ could be considered zero. Due to the high air relative humidity and low the vapor pressure deficit (Figure 5) during the period of horizontal precipitation, the water vapor concentration gradient between soil/plant and air was so low that the condition failed to reach the latent heat status [29,30]. Thus, the evapotranspiration barely occurred and could be ignored in this period. Therefore, the value of $P_h$ was calculated by the mass increase of the lysimeter; the calculation formula was equation (6). The data was recorded at 10-minute intervals.

$$P_h = \Delta W \tag{6}$$

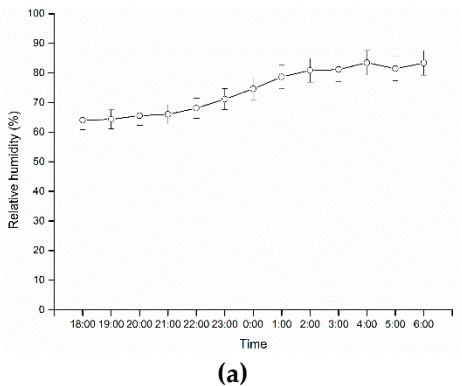
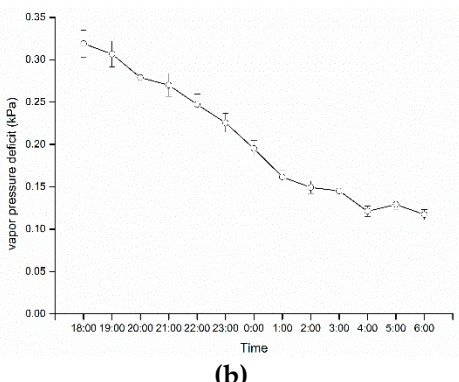

(a)　　　　　　　　　　　　　　　　　　　　　(b)

**Figure 5.** The variation of relative humidity and vapor pressure deficit during typical night ((**a**) the variation of relative humidity; (**b**) the variation of vapor pressure deficit).

## 2.4. Meteorological Data Measurements

Meteorological data, which included precipitation ($P$), net radiation ($R_n$), air temperature ($T_a$), dew point temperature ($T_d$), relative humidity of air ($RH_a$) and wind speed ($WS$), were recorded synchronously by the HOBO automatic small weather station (U30, Onset, USA). The variation of precipitation during total experimental period was as shown in Figure 6. The station was located in the center of the four lysimeters, and was installed 1.2 m (below the canopy) and 15 m (above the canopy) height above from ground (Figure 3). There was no shelter around the station, and it was well ventilated. The canopy temperature ($T_c$) was measured using temperature sensors (ST-2955, SINTEK, USA). In order to measure the soil water content ($SWC$), water potential ($\varphi_s$) and temperature of the top soil ($T_s$), soil water sensors (ECH$_2$O-TE System, Decagon Devices, USA) were embedded into the soil inside each of the four lysimeters at 0-5cm below the ground surface. The data were recorded at 10-minute intervals. The leaf area index was measured using the canopy analyzer meter (LAI-2200C, LI-COR, USA) and recorded three times per month.

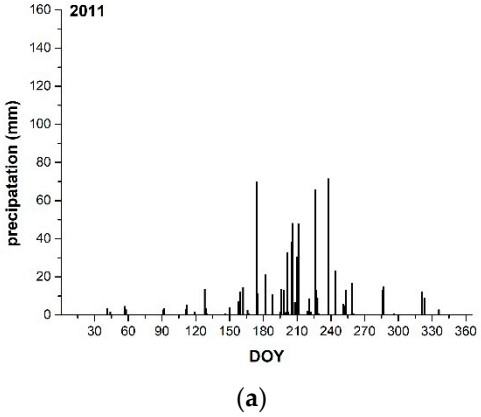
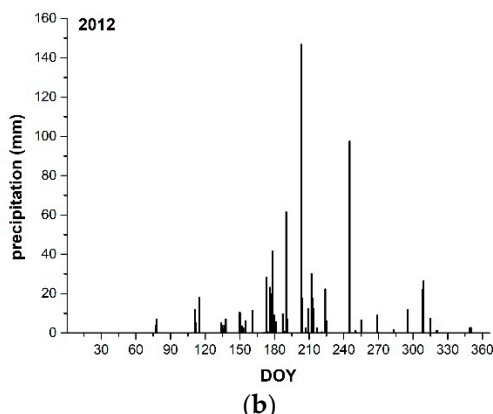

(a)　　　　　　　　　　　　　　　　　　　　　(b)

**Figure 6.** The variation of precipitation during total experimental period ((**a**) the variation of precipitation in 2011; (**b**) the variation of precipitation in 2012).

## 2.5. Data Analysis

The data used in this study were selected from the measurements which met the following conditions: (1) long-wave radiation must be greater than short-wave radiation, i.e., net radiation must be zero or less [31]; (2) the average wind speed must be less than 0.5 m/s; and (3) the weather must be sunny or with few clouds (cloud amount < 1/10 of sky cover) [32].

SPSS software version 18.0 was used for statistical analyses of the measured duration and quantity of horizontal precipitation data. The Origin software version 9.0 (OriginLab) was used to

analyze the relationship between the quantity of horizontal precipitation and meteorological factors. The significance of the relationship was tested using one-way ANOVA with $p$-values = 0.05.

## 3. Results

### 3.1. The Duration and Quantity of Horizontal Precipitation

A typical night (2011.12.2–2011.12.3; mean *WS* was 0.37 m/s; *RH* was 55.4%) was selected to describe the process of horizontal precipitation (Figure 7). Hourly distribution of the horizontal precipitation was between 0 and 0.1 mm and the total of the horizontal precipitation per night was between 0.2 and 0.4 mm for the four tested plant species. The amount of horizontal precipitation was higher in grass (*Lolium perenne*) (0.38 mm) than in other tree species, and higher in broad-leaved tree species (*Q. variabilis*) than in conifers tree species.

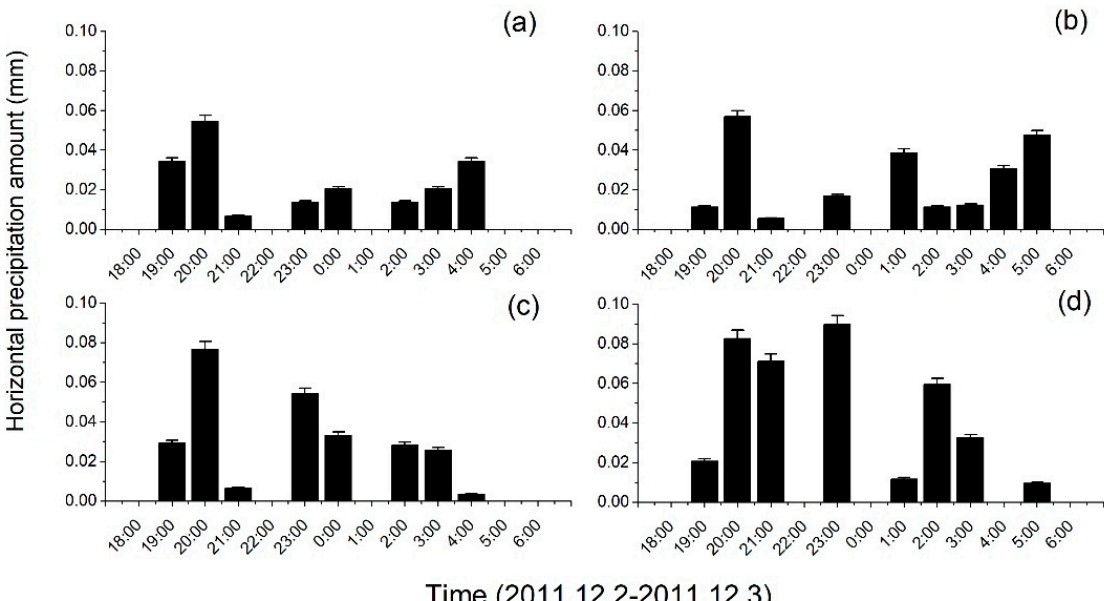

**Figure 7.** The duration of horizontal precipitation a typical night ((**a**) *Pinus tabulaeformis*; (**b**) *Platycladus orientalis*; (**c**) *Quercus variabilis*; (**d**) *grass (Lolium perenne)*; 2–3 December 2011).

Rainy days with horizontal precipitation were more frequent during the non-growth season (i.e., September–February) than in the growth season (March–August) in 2011 and 2012 (Figure 8). In the non-growth season of 2011, the recorded times when horizontal precipitation occurred were 116d, 117d, 149d and 152d in *Pinus tabulaeformis*, *Platycladus orientalis*, *Quercus variabilis* and *grass*, respectively, accounting for about 64.1%, 64.6%, 82.3% and 84.0% of the whole dates in the non-growth season. In 2012, the corresponding times were 98d, 143d, 164d and 159d, accounting for about 53.4%, 78.6%, 90.1% and 87.4% of the non-growth season in the four plant species, respectively. At the same time, the number of days of horizontal precipitation at the monthly level in non-growth season accounted for about 82% to 85% and 20% to 25% in growth season. There was a significant difference in the number of horizontal precipitation days between the *grass* and *Platycladus orientalis* and *Pinus tabulaeformis* ($p < 0.05$). There were also significant differences between *Quercus variabilis* and *Platycladus orientalis* and *Pinus tabulaeformis* ($p < 0.05$). Furthermore, there were more days of horizontal precipitation for the *grass* than for the other three species.

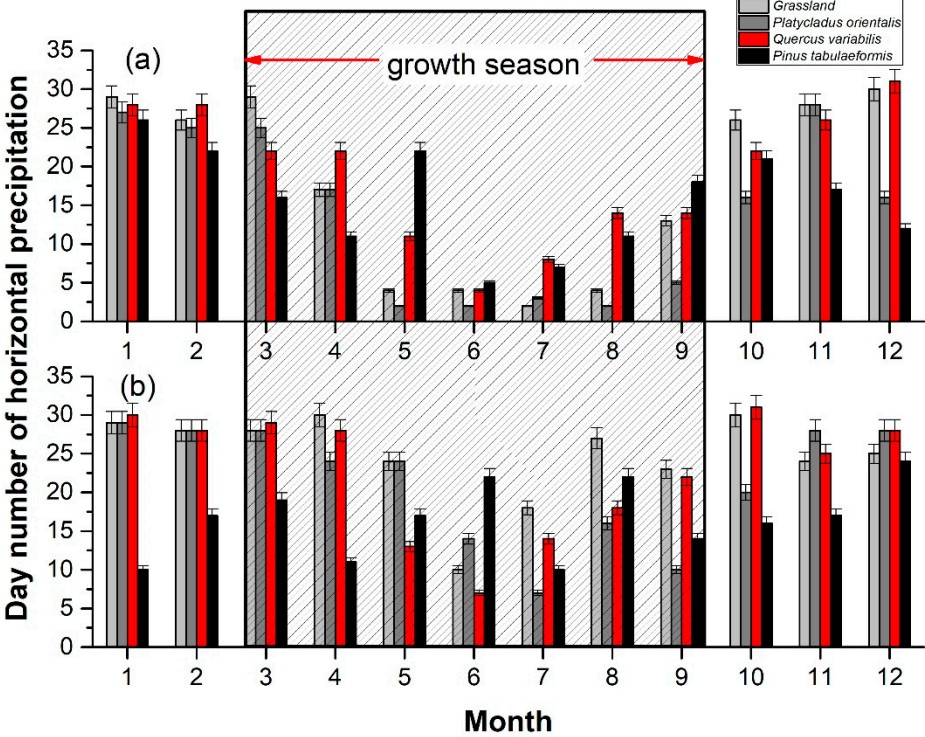

**Figure 8.** Monthly horizontal precipitation duration for four forest lands from 1 January, 2011 to 31 December, 2012 ((**a**) was 2011 and (**b**) was 2012). Data were calculated by weight changes, calculated using a weighing lysimeter system.

Figure 9 shows variations in the monthly horizontal precipitation quantities in 2011 and 2012. The overall trend throughout the year was the same as that of the horizontal precipitation duration of each month. The average monthly horizontal precipitation was 4.5 mm in the non-growing season, while it was a mere 1.6 mm in the growing season. Horizontal precipitation peaked in Dec. or Jan., while the minimum value appeared from June to July. The total amount of horizontal precipitation in the year was about 33 mm. The horizontal precipitation quantity in 2011 and 2012 accounted for 4.61% and 4.23% respectively of the annual precipitation. The horizontal precipitation amounts for *Pinus tabulaeformis*, *Platycladus orientalis*, *Quercus variabilis* and *grass* in 2011 were 30.0, 26.2, 37.9 and 38.7 mm, respectively, which accounted for 4.16%, 3.64%, 5.26% and 5.37% of the annual precipitation. In 2012, the horizontal precipitation amounts of four forest lands were 30.47, 24.39, 35.25 and 41.06 mm, respectively, which accounted for 3.92%, 3.15%, 4.55% and 5.29% of the annual precipitation. Via analysis of variance, there were significant differences of monthly horizontal precipitation amounts between *Platycladus orientalis* and grass (*Lolium perenne*), *Quercus variabilis*. By quantitative analysis, the grass's horizontal precipitation amounts, which reached about 5 mm on average in November and December, were more than those of the other three forest lands.

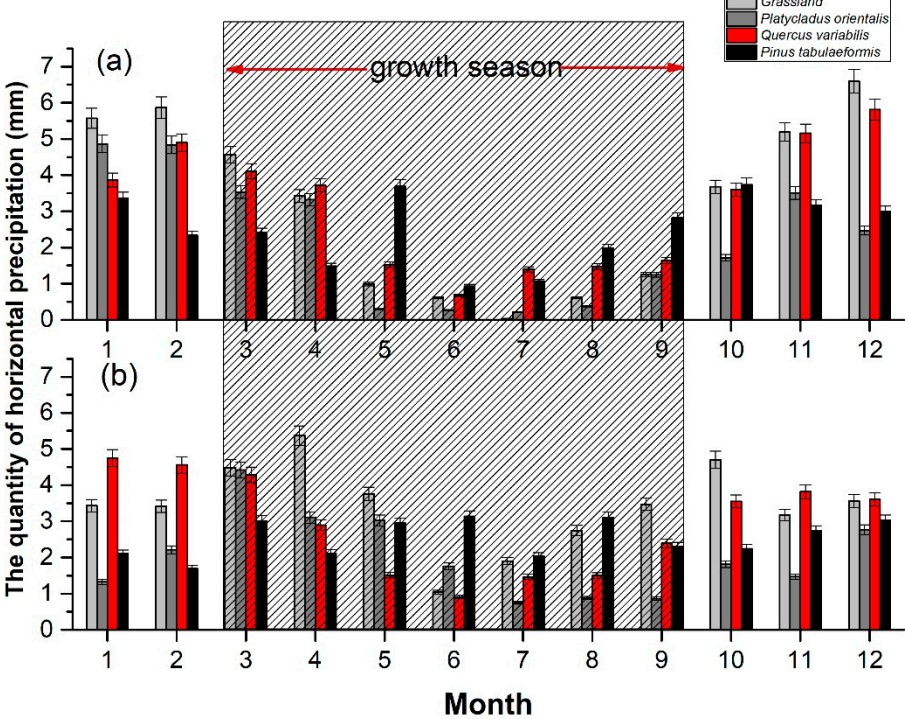

**Figure 9.** Monthly horizontal precipitation quantity for four forest lands from 1 January 2011 to 31 December 2012 ((**a**) was 2011 and (**b**) was 2012). Data were calculated by weight changes, calculated using a weighing lysimeter system.

### 3.2. Canopy Condensation and Water Vapor Absorption by Soil

When the weight of the lysimeter increased, the following conditions could be used to identify whether that increase was is caused by canopy and soil condensation or by water vapor absorbed by soil.

(1) When the air temperature ($T_a$) was lower than the dew point temperature ($T_d$), and the temperature of the canopy ($T_c$) and soil ($T_s$) were lower than the air temperature ($T_a$), the weight increase was due to water vapor condensation in the canopy and soil.

(2) When the air temperature ($T_a$) was higher than the dew point temperature ($T_d$), and the water potential of air ($\varphi_a$) was higher than that of the soil ($\varphi_s$), the weight increase was due to water vapor absorption by soil.

The relationship between the quantity of water vapor condensation and different kinds of temperature is shown in Figure 10. At night time, water vapor condensation produced a period of $T_d > T_a > T_s$ and $T_c$. The quantity of water vapor condensation was 0.07 mm, accounting for 31.81% of the total quantity of horizontal precipitation (0.22 mm). When the period of $T_d < T_a$ came, water vapor absorption by soil was controlled by the gradient of water potential (Figure 11). The variation of air and soil water potential was opposite, and the gradient of water potential decreased.

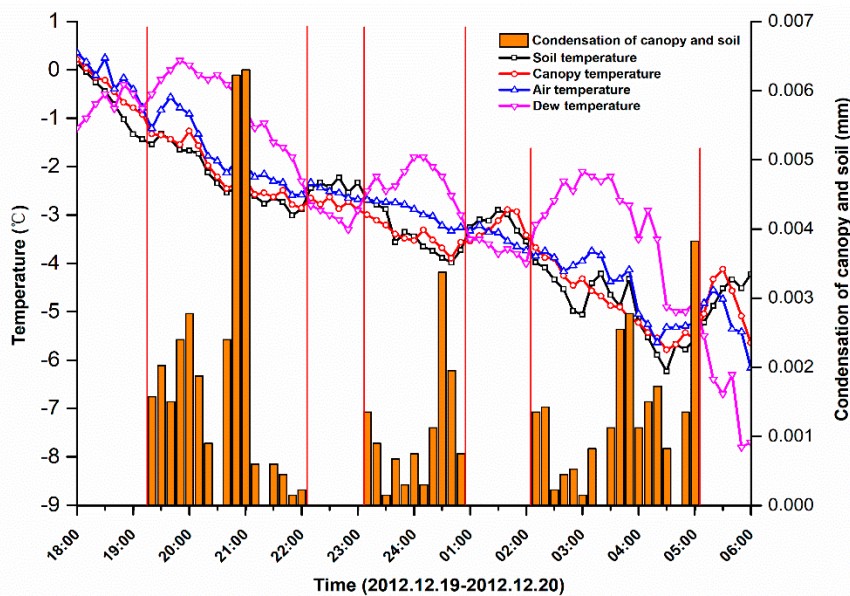

**Figure 10.** Changes of condensation vs different kinds of temperature on a typical night (including air, soil, canopy and dew temperature; 19–20 December 2012).

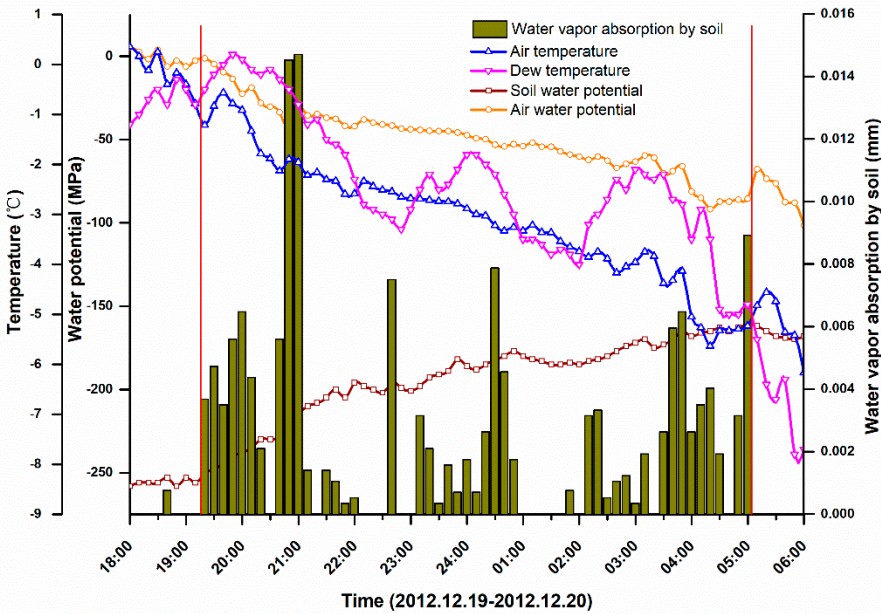

**Figure 11.** Changes of water vapor absorption by soil vs. different kinds of water potential on a typical night (including air and soil water potential; 19–20 December 2012).

In accordance with the aforementioned conditions, the canopy and soil condensation and water vapor absorbed by soil were analyzed for *P. orientalis* (Figure 12). Since there was little variation of the leaf area index throughout the year, analyses were carried out for 30 days in June and 31 days in December, 2011; these two months represented the growing and the non-growing seasons, respectively. The results showed that the quantity of canopy and soil condensation and water vapor absorbed by soil in the non-growing season were all higher than those in the growing season. As shown in Figure 12a, in December 2011, the total quantity of water vapor absorbed by the soil was 4.88 mm over 31 days, which was 2.85 times that of the canopy and soil condensation (1.71 mm over 29 days); however, as shown in Figure 12b, the results were the opposite in June, at which time the quantity of canopy and soil condensation was 0.41 mm over 4 days, which was 2.05 times that of the water vapor absorbed by soil (0.2 mm over 4 days). For the variation during each day in December, the quantity of water

vapor absorbed by the soil was 0.16 mm, which was 2.67 times that of canopy and soil condensation (0.06 mm). In June, the quantity of canopy and soil condensation was 0.1 mm which was 2 times that of the water vapor absorbed by the soil (0.05 mm). Thus, the water vapor absorbed by the soil was greater than canopy and soil condensation, not only in terms of frequency, but also in terms of cumulated quantity in the non-growing season. However, in the growing season, the opposite occurred.

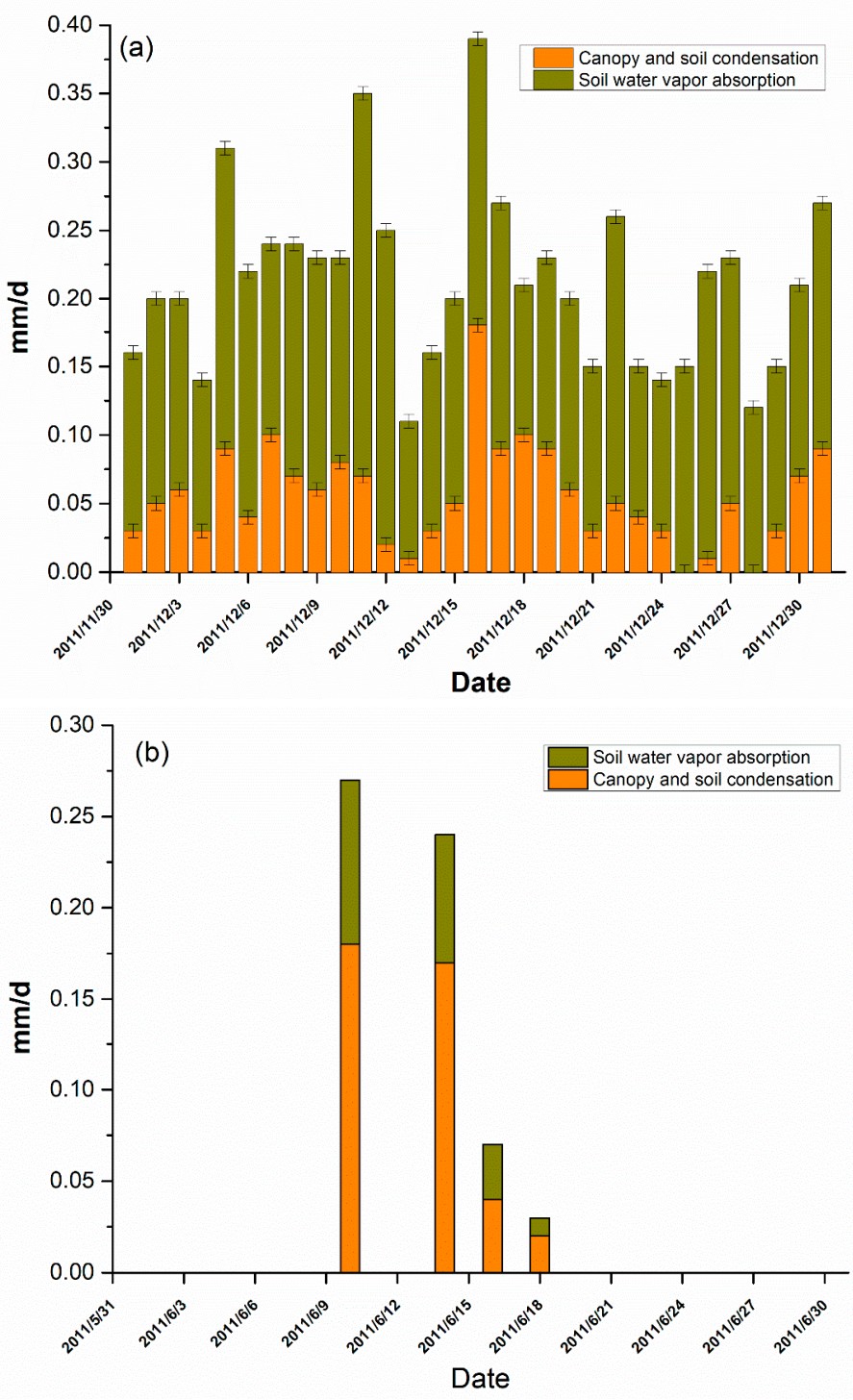

**Figure 12.** Quantitative variations among soil water vapor absorption and canopy and soil condensation on the basis of the measured soil water content at 0–5 cm, soil surface temperature, and calculated dew point temperature. Daily data was recorded from 1–30 June (**b**) and from 1–31 December (**a**).

### 3.3. The Relationship between Horizontal Precipitation and Meteorological Factors

The relationship between horizontal precipitation and meteorological factors is shown in Table 3. Air temperature, soil temperature and wind speed are negatively correlated with the quantity and duration of horizontal precipitation. However, soil temperature was shown to be highly significant on the quantity of horizontal precipitation ($p < 0.01$), which was the same as wind speed. Relative humidity and atmospheric pressure had a low effect on the quantity of horizontal precipitation. In terms of composition, air temperature, soil temperature and wind speed are negatively correlated with the quantity and canopy and soil condensation and were highly significant for the quantity of horizontal precipitation ($p < 0.05$), which was negatively correlated with air temperature (r = −0.152) and wind speed (r = −0.115). For soil water vapor absorption, however, air temperature was highly correlated with soil temperature ($p < 0.05$).

**Table 3.** The relationship between horizontal precipitation and meteorological factors.

| Size | | Air Temperature (°C) | Soil Temperature (5 cm) (°C) | Relative Humidity (%) | Wind Speed (m/s) | Atmospheric Pressure (hPa) |
|---|---|---|---|---|---|---|
| HP quantity over day time scale | Pearson correlation | −0.108 | −0.245 ** | 0.089 | −0.128 * | 0.071 |
| | *p* value | 0.147 | 0.000 | 0.239 | 0.09 | 0.346 |
| | N | 176 | 176 | 176 | 176 | 176 |
| Canopy and soil condensation quantity over day time scale | Pearson correlation | −0.152 * | −0.138 * | 0.079 | −0.115 * | 0.05 |
| | *p* value | 0.035 | 0.424 | 0.295 | 0.046 | 0.508 |
| | N | 176 | 176 | 176 | 176 | 176 |
| Soil water vapor absorption quantity over day time scale | Pearson correlation | −0.034 | −0.173 * | 0.007 | −0.022 | 0.011 |
| | *p* value | 0.077 | 0.033 | 0.932 | 0.706 | 0.885 |
| | N | 176 | 176 | 176 | 176 | 176 |
| HP duration over day time scale | Pearson correlation | −0.140 * | −0.220 ** | 0.130 | −0.275 ** | −0.00 5 |
| | *p* value | 0.003 | 0.005 | 0.372 | 0.807 | 0.977 |
| | N | 31 | 31 | 31 | 31 | 31 |

\* Correlation is significant at the 0.05 level (2-tailed). \*\* Correlation is significant at the 0.01 level (2-tailed).

## 4. Discussion

### 4.1. Forestland-Specific Differences in Horizontal Precipitation

The four investigated forestlands showed general differences in horizontal precipitation. In the growth season, the horizontal precipitation duration of *Quercus variabilis* (broad-leaved tree) was longer than those of *Platycladus orientalis*'s and *Pinus tabulaeformis*'s (coniferous tree). This may be due to the leaf area index (LAI) [5]. According to the measurements, the average LAI of *Q. variabilis*, *P.orientalis* and *P.tabulaeformis* in the growth season were 3.32, 2.98 and 2.43, respectively. Higher LAI made the leaf surface more likely to condense water vapor. In the non-growth season, leaf loss occurred in *Q. variabilis,* which was a temperate deciduous forest. Dead branches and leaves were shed, covering the soil [3]. As a result, the radiation, temperature and soil evaporation received from the ground were changed. The temperature on the ground decreased, becoming closer to the temperature of the dew point. Thus, it was more likely to precipitate horizontal precipitation at night. However, there were few dead branches and leaves on the ground under *P. orientalis* and *P. tabulaeformis* in the non-growth season. The temperature of the soil adherent to the stratum was higher than that of *Q. variabilis*, so the duration of horizontal precipitation was less. Pan also indicated that temperature was the main reason for horizontal precipitation [32].

The amount of horizontal precipitation in grass (*Lolium perenne*) was more than that of other species. There were three possible reasons: (1) the degree of coverage of grass, 85%, was higher than

that of other forestlands that were 2500 plants/hm$^2$. Surface grass coverage increasing the contact area between air and vegetation, thereby improving horizontal precipitation [33]; (2) the surface temperature of grass (*Lolium perenne*) was closer to the dew point [31]. The temperature of the surface and at 5, 10, 30, 100 and 200 cm above the ground was measured. There was a significant gradient at substratum air temperature below 30 cm. The closer it was to the ground, the lower the temperature was, and therefore, the closer it was to the dew point; (3) the soil in the grass (*Lolium perenne*) was drier than that of the other three forest lands. Not only did the water vapor precipitate onto plants, but it was also absorbed by the soil.

### 4.2. The Effect of Horizontal Precipitation on Water Cycle Changing

From our study, we concluded that horizontal precipitation, which accounted for around 4.61% of annual rainfall, was a significant water source in the mountainous area of northern China. Gao [34] found that dew accounted for 2.28% to 6.67% of the precipitation in the Heihe river in China, which was consistent with our findings. Horizontal precipitation added, along with rain and snow, to the ecosystem's water input. Furthermore, these two forms, i.e., water vapor absorbed by the soil and canopy and soil condensation, and horizontal precipitation, changed the water cycle. Thus, different types of land use could have an impact on the two forms of horizontal precipitation, thereby changing sediment delivery [17]. The water vapor absorbed by the soil may change the soil structure and moisture levels, which could affect soil erosion on a large scale. Van [18] and Cerdà also observed similar consequences in a different region [19].

### 4.3. The Meteorological Factors that Affect Horizontal Precipitation

Soil temperature was a key factor affecting the quantity of water vapor absorbed by soil. The water content of top soil was an important factor for water absorption by soil for horizontal precipitation. Since wind could easily affect leaf surfaces, it was difficult to produce horizontal precipitation data for leaf surfaces. Some earlier studies indicated that the formation of horizontal precipitation benefited from breeze [22,35]. In fact, the effect of wind on horizontal precipitation formation is complicated. Wind could enhance the movement of water vapor and heat, both in horizontal and vertical directions [36]. Furthermore, wind was a main factor for vapor diffusion; thus the probability of dew deposit increased. When wind was weak or close to zero, a boundary layer formed [37]. The water molecules diffused over this layer and randomly hit the droplet into which they were incorporated [10]. Then, a concentration gradient of water molecules formed around the droplet, and the quantity of horizontal precipitation might accumulate [38]. As such, the relationship between horizontal precipitation and wind in the present study represented the integrated influence of wind.

Wind could also affect the lysimeter weight because strong wind in a vertical direction could cause changes in mass [3]. The response of horizontal precipitation to wind is a complex process. Turbulence transported water vapor towards the canopy and soil surfaces and made temperaturse decrease quickly to dew and frost point temperatures [39]. Therefore, horizontal precipitation was more abundant when wind was mild. But for strong wind, it may cause water vapor diffusion and hinder the formation of horizontal precipitation.

## 5. Conclusions

The total amount of horizontal precipitation in the year was about 33 mm, which represented about 4.61% and 4.23% of the annual precipitation in 2011 and 2012 respectively. Hence, the horizontal precipitation was a significant water source that could affect the water cycle of forest land in the mountainous area in northern China. At the same time, the day number of monthly values in the non-growth season accounted for about 82% to 85%. There were two forms of horizontal precipitation, namely, water vapor absorbed by the soil and canopy and soil condensation. During the non-growing season, water vapor absorbed by the soil was greater than canopy and soil condensation, not only in

terms of frequency, but also in cumulated quantity. Air temperature, soil temperature and wind speed were negatively correlated with the quantity and duration of horizontal precipitation.

**Author Contributions:** J.J. and W.L. conducted field experiment, performed data analysis, and wrote the draft manuscript. W.Y. and X.C. conceived the study, designed the experiment. All authors contributed to discussion and interpretation of resulting data.

**Funding:** The research project on which this paper is based was funded by the China Natural Science Foundation (No. 41807162). All experiments comply with the current laws of the country in which they were performed. All data necessary to replicate the findings presented in the study are available in the paper. These data could be available upon request to all interested researchers.

**Conflicts of Interest:** The authors declare no conflict of interest.

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
