# Peer review of "Characteristics of Horizontal Precipitation in Semi-Humid Forestland in Northern China"

_water, doi:10.3390/w11050975_

Round 1
Reviewer 1 Report
General comments:
This study evaluated the quantity and duration of horizontal precipitation for four forests in semi-humid climate region. The experimental design using large lysimeters is interesting and I think this study provides precious data on horizontal precipitation. However, the manuscript is not well-organized and I don’t understand what novelty and/or findings of this study are. The authors should present what the scientific identity of the study is when compared to other studies on horizontal precipitation throughout the world.
The authors should follow the authors’ guideline. For example, the authors should cite literature in numerical sequence in the text. Furthermore, in the abstract section, they shouldn’t show the headings, such as “Background”.
Substantial revisions are needed before reconsidering publication in the Water.
Specific comments:
Introduction
I don’t understand the research gap of this study. The objective of this study is also vague. The authors should make them clear in this section.
Methods
I don’t understand the observation period and the time interval of data acquisition.
The authors should present time series of precipitation, i.e., rainfall and snow, during the observation period.
The authors describe that evapotranspiration could be ignored in the water balance equation, but I don’t believe it. They should present time series of evapotranspiration during the observation period in the Results section.
Results
I don’t find Figure 8b anywhere.
Discussion
The authors try to explain mechanisms behind the results obtained while citing the knowledge the previous other studies have found, but there’s no cited literature in this section.
The authors first show the correlation coefficient between water content and air temperature or wind speed, but they shouldn’t show the results in this section. They should first show them in the Results section.
The authors should show findings and/or novelty and present the scientific identity of this study in this section.
The heading, “Discussion” is an uncountable noun.
Conclusions
This section doesn’t show the conclusions, but only the results.
Author Response
Dear Reviewer:
Thank you for your letter and for the comments concerning our manuscript entitled “Characteristics of horizontal precipitation in semi-humid forestland in northern China”. I am sorry that I wrote back after such a long time. Those comments are all valuable and very helpful for revising and improving our paper, as well as the important guiding significance to our researches. We have studied comments carefully and have made correction which we hope meet with approval. Revised portion are marked in the paper(word file). The main corrections in the paper and the responds to the reviewer’s comments are as flowing:
General comments:
1. Q: The manuscript is not well-organized and I don’t understand what novelty and/or findings of this study are.
A: The manuscript has been revised for whole parts. At the same time, the novelty and findings of this study has been given in the section of discussion.
2. Q: The authors should present what the scientific identity of the study is when compared to other studies on horizontal precipitation throughout the world.
A: The scientific identity of the study has been presented in the section of discussion.
3: Q: The authors should follow the authors’ guideline. For example, the authors should cite literature in numerical sequence in the text. Furthermore, in the abstract section, they shouldn’t show the headings, such as “Background”.
A: We have followed the authors’ guideline. We have cited literature in numerical sequence in the text. And the abstract has been revised.
Specific comments:
1.Q: Introduction: I don’t understand the research gap of this study. The objective of this study is also vague. The authors should make them clear in this section.
A: Introduction has been rewritten, which included the research gap and objective. The specific research gap was in paragraph 2 of Introduction and the objective of this study was in the paragraph 4 of Introduction.
2.Q: Methods: I don’t understand the observation period and the time interval of data acquisition
A: The observation period and the time interval of data acquisition have been added in section 2.3
3.Q: Methods: The authors should present time series of precipitation, i.e., rainfall and snow, during the observation period.
A: The time series of precipitation has been presented as the figure 5 showing,
4.Q: Methods: The authors describe that evapotranspiration could be ignored in the water balance equation, but I don’t believe it. They should present time series of evapotranspiration during the observation period in the Results section.
A: We have listed the statistic of relative humidity and vapor pressure deficit and explain the mechanism of latent heat. Besides, we have cited relative literatures about little evapotranspiration. According all these, we could proof that the evapotranspiration could be ignored in the period of horizontal precipitation.
5.Q: Results: I don’t find Figure 8b anywhere.
A: I’m very sorry and the figure has been inserted.
6:Q: Discussion: The authors try to explain mechanisms behind the results obtained while citing the knowledge the previous other studies have found, but there’s no cited literature in this section.
A: The necessary literatures has been supplied.
7:Q: Discussion: The authors first show the correlation coefficient between water content and air temperature or wind speed, but they shouldn’t show the results in this section. They should first show them in the Results section
A: The correlation coefficient between water content and air temperature or wind speed has been showed in the results part.
8:Q: Discussion: The authors should show findings and/or novelty and present the scientific identity of this study in this section.
A: The findings and novelty and present the scientific identity of this study have been showed in this section.
9: Q: Discussion: The heading, “Discussion” is an uncountable noun.
A: The Discussion has been revised.
10: Q: Conclusions: This section doesn’t show the conclusions, but only the results
A: This section has been rewritten, and it has showed the conclusions.

Reviewer 2 Report
Please, see the notes in the attached document

Author Response
Dear Reviewer:
Thank you for your letter and for the comments concerning our manuscript entitled “Characteristics of horizontal precipitation in semi-humid forestland in northern China”. I am sorry that I wrote back after such a long time. Those comments are all valuable and very helpful for revising and improving our paper, as well as the important guiding significance to our researches. We have studied comments carefully and have made correction which we hope meet with approval. Revised portion are marked in the paper(word file). The main corrections in the paper and the responds to the reviewer’s comments are as flowing:
Specific comments:
1. Q: L52 remove space
A: It has been modified
2. Q: L64 add a space after the table
A: It has been modified
3. Q: L98 Why? Please explain
A:The amount of precipitation in Beijing mountains is less than 400mm, so it belongs to semi-humid climate region. And due to the adjust of literature structure, the sentence has been delete.
4. Q: L108 year?
A: This literature has been inserted
5. Q: L152 Replace de , by .
A: It has been modified
6. Q: L153 Please, add some information of the wind (speed and direction). Please, add some information of the exposition of the slopes
A: These content have been inserted in this paragraph (Section 2.2)
7. Q: L153 what about scrubs? species? higth? density per m2?
- Grassland? higth and density per m2.
-what about soil texture?
A: These content have been inserted in this paragraph
8. Q: L153 Please, for all the non chinese readers, add a map of china to locate the Beijing area.
-Please, to respect the international conventions, use brown scale colors for topography.
-Please, for all the non chinese readers, add some photos (Pinus, Quercus, Platycladus, ...) of the forest land in semi-humid of the Research site.
-Please, add a solar exposure map of the study area.
-Please, add a map with vegetation tipes: forest, scrub, grassland.
A: The map of china to locate the Beijing area has been inserted (Figure 1) and use brown scale colors for it; The detail of forest photos, solar exposure map and vegetation types map referred to Shi (2010)
9. Q: L158 Please, add photo.
A: The photo has been inserted (Figure 3)
10. Q: L203 few clouds? 0/8? 1/8? 2/8? of sky cover
A: These content have been inserted in this paragraph (cloud amount < 1/10 of sky cover)
11. Q: L323 of?
A: It has been modified

Reviewer 3 Report
Dear Authors,
Please specify which grass species was a dominant in your research. Tree species used in experiment were correctly defined except grass, which in some cases were even named as 'grassland'. So, was this only one species (grass) or some of them (grassland) ? It should be clearly defined.
Author Response
Dear Reviewer:
Thank you for your letter and for the comments concerning our manuscript entitled “Characteristics of horizontal precipitation in semi-humid forestland in northern China”. I am sorry that I wrote back after such a long time. Those comments are all valuable and very helpful for revising and improving our paper, as well as the important guiding significance to our researches. We have studied comments carefully and have made correction which we hope meet with approval. Revised portion are marked in the paper(word file). The main corrections in the paper and the responds to the reviewer’s comments are as flowing:
Specific comments:
Q: Please specify which grass species was a dominant in your research. Tree species used in experiment were correctly defined except grass, which in some cases were even named as 'grassland'. So, was this only one species (grass) or some of them (grassland) ? It should be clearly defined.
A: The grass species was Lolium perenne that was a dominant in our research. It was the only one species that forms grass. These content have been inserted in section 2.2

Round 2
Reviewer 2 Report
The authors responded satisfactorily to almost all the issues raised by the reviewers and substantially improved the article.
In order to improve the paper, some corrections have to be made:
“The map of china to locate the Beijing area has been inserted (Figure 1) and use brown scale colors for it“
- Figure 1 must be improved (e.g. lack quality and legibility and the legend is missing)
- brown scale colors - in the version I received to review, I only see gray!
“The detail of forest photos, solar exposure map and vegetation types map referred to Shi (2010)”
- It is necessary to be inserted in the paper and not just refer to a publication of very difficult access
So, although it presents some shortcomings, due to its originality, I am of the opinion that it is in the interest of many researchers and therefore deserves to be published after a new revision.
Author Response
Dear Reviewers:
Thank you for your letter and for the comments concerning our manuscript entitled “Characteristics of horizontal precipitation in semi-humid forestland in northern China”. Those comments are all valuable and very helpful for revising and improving our paper, as well as the important guiding significance to our researches. We have studied comments carefully and have made correction which we hope meet with approval. Revised portion are marked in the paper. The main corrections in the paper and the responds to the reviewer’s comments are as flowing:
Q1: “The map of china to locate the Beijing area has been inserted (Figure 1) and use brown scale colors for it“
- Figure 1 must be improved (e.g. lack quality and legibility and the legend is missing)
- brown scale colors - in the version I received to review, I only see gray!
A: Figure 1 has been rewritten and inserted in the paper
Q2: “The detail of forest photos, solar exposure map and vegetation types map referred to Shi (2010)”
- It is necessary to be inserted in the paper and not just refer to a publication of very difficult access
A: I am so sorry that I did not modify it as required, and this time forest photos and vegetation types map have been inserted in Figure 2. However, we don’t have enough data to plot the solar exposure map. I felt so sorry and hoped for your understand. Thank you.
